# AFTERSTATE REINFORCEMENT LEARNING FOR CONTINUOUS CONTROL

## ABSTRACT

Humans consider the consequence of taking action in decision-making. In particular, we imagine what will happen upon executing an option of interest. In actor-critic algorithms, the critic evaluates actions from the actor by explicitly taking the action representation as input whereas the conventional value-based methods such as Deep Q-Network (Mnih et al., 2015) do not explicitly deal with such action information. With the action being input, the critic's task in the actor-critic framework can be decomposed as follows; (I) learning the utility of action on the environment, (II) learning the future consequence of the action. Our work aims to enhance the critic's imagination (I) by utilising the environment model based on the model-based RL framework. To this end, our key insight is that all actions landing on the same next state are equivalent. In continuous action space tasks, robot control and painting, we show the efficacy of our method.

## 1 INTRODUCTION

Solving complex tasks from the high-dimensional, sensory input is an interesting yet challenging problem with many real-world applications and Actor-Critic algorithms are a powerful common choice to perform tasks (Lillicrap et al., 2015; Schulman et al., 2017; 2015). It is a family of methods that are two time-scale algorithms in which the critic uses temporal-difference learning to learn the future consequence and the actor is updated in an approximate gradient direction based on the evaluated action (or query point) by the critic. Recently, a significant amount of work has been made to improve the actor; with a sophisticated loss formulation to encourage the diversity (Haarnoja et al., 2018), a two-phase framework that allows the actor to select more than one action and refine the actions to make decisions (Kalashnikov et al., 2018; Dulac-Arnold et al., 2015).

In actor-critic algorithms, the critic evaluates actions from the actor by explicitly taking the action representation as input whereas the conventional value-based methods such as Deep Q-Network (Mnih et al., 2015) do not explicitly deal with such action information. Indeed, action representation is useful to convey what kind of action the agent is taking and has been shown to help the agent generalize over unseen actions (Jain et al., 2020). With the action representation being an input, now the critic needs to learn to explicitly process the information of action. But what does this mean? Humans often picture the state after executing the action rather than thinking about the utility of action in the process of decision-making. Suppose, in the game of Tic-tac-toe, we are imagining how the opponent behaves a few steps ahead to decide on your next move. In this planning process, we think about what the opponent will do if you place your next marker on a specific position on the board. Thus, we are thinking more about the state after the execution of the action. Similarly, the critic's task in the actor-critic framework can be decomposed as follows; (I) learning the utility of action on the environment, (II) learning the future consequence of the action.

Afterstate Sutton & Barto (2018) (or Poststate in Powell (2007)) implements the intuition of our planning process in the value-based methods by evaluating the state after the action has been executed, called *afterstate*. And, the concept has been applied to some applications, the boardgames (Szubert & Jaśkowski, 2014; Matsuzaki, 2021; Antonoglou et al., 2021), painting (Huang et al., 2019; Singh & Zheng, 2021; Xu & Zhang; Singh et al., 2021; 2022), or Power-supplying network management (Yoon et al., 2021). Prior works, however, either build on the value-based method for discrete tasks or assume the knowledge of the environment (Huang et al., 2019), e.g., the state transition or the reward of the unexecuted action.

The model-based framework offers the capability to learn models approximating the environment, e.g., the state transition or the reward function. So, we propose to combine the model-based and aftertaste frameworks to lift the task of learning the utility of action on the environment to the dynamics model and let the critic focus on learning the future value of the action. To evaluate the efficacy of our method, we conduct experiments in three different continuous control tasks. Our results demonstrate that our approach outperforms the conventional actor-critic algorithms in the tasks by delegating the heaviness of understanding the action information. Our primary contribution is (i) introducing the heaviness of the critic's issue that it needs to learn the action utility, and (ii) implementing our intuition into the critic in a principled way to approach the issue. And we empirically show that our proposed architecture, the afterstate model-based RL, enables efficient decision-making in continuous control tasks.

## 2 PROBLEM SETUP

We consider a standard reinforcement learning setup consisting of an agent interacting with an environment in discrete timesteps. Specifically, we consider a Markov Decision Process (MDP), defined by a tuple $\{\mathcal{S}, \mathbb{A}, \mathcal{T}, \mathcal{R}, \gamma\}$ of states, actions, transition probability, reward function, and a discount factor, respectively. When the action space $\mathbb{A}$ is continuous, it has a $D$-dimensional continuous parameterization, $c_a \in \mathbb{R}^D$. This kind of continuous parameterization of action space has been applied to other domains, e.g., item-set in recommender systems where each item has $D$-dimensional representations encoding product-specific information (Chen et al., 2019; Jain et al., 2020). At each timestep $t$ the agent receives an observation $s_t$, selects an action $a_t$ then receives a reward $r_t$ as well as the next observation $s_{t+1}$. Thus, the objective of the agent is to learn a policy $\pi(a|s)$ that maximizes the expected discounted reward over evaluation episodes, $\mathbb{E}_\pi \left[ \sum_t \gamma^{t-1} r_t \right]$.

In actor-critic methods, the policy $\pi_\phi$ (also known as the actor) is learning from the critic ($Q(s,a)$ evaluates action $a$ in state $s$) via the deterministic policy gradient theorem (Silver et al., 2014):

$$\nabla_\phi J(\phi) = \mathbb{E}_{s \sim p_\pi}[\nabla_a Q^\pi(s,a)|_{a=\pi_\phi(s)} \nabla_\phi \pi_\phi(s)]$$

where $Q^\pi(s,a) = \mathbb{E}_{s_i \sim p_\pi, a_i \sim \pi}[R_t|s,a]$, the expected return given action $a$ in state $s$ and following the policy $\pi$ after.

The critic is essentially different from the conventional state-action value-based methods (Mnih et al., 2015) as it is processing the action representation directly. Thus, the critic's task can be decomposed as follows; (I) learning the utility of action on the environment, (II) learning the future consequence of the action.

## 3 PROPOSITION: MODEL-BASED AFTERSTATE RL

In this section, we present the concept of the Afterstate framework and explain how we incorporate this into the critic in the actor-critic framework.

### 3.1 AFTERSTATE FRAMEWORK FOR REINFORCEMENT LEARNING

By definition, using the state-value (V) the Q-value can be expanded as follows;

$$Q(s_t, a_t) = r(s_t, a_t) + \gamma * V(s_{t+1}) \tag{1}$$

where $V(s_{t+1}) = \sum_{a \in A} \pi(a_{t+1}|s_{t+1}) Q(s_{t+1}, a_{t+1})$. Here, we assume that the environment is deterministic so we have no expectation over $s_{t+1}$. In the supplementary material (Sec.2) we discuss the limitation of this assumption in the aftertaste framework. Thus, the temporal difference update can be formulated as $V(s) \leftarrow V(s_t) + \alpha(r(s_{t+1}, a_{t+1}) + \gamma V(s_{t+2}) - V(s_t))$ where $\alpha$ is the learning rate and $\gamma$ is the discounting factor. In this formulation, we need the knowledge of future time-steps, e.g., $r(s_{t+1}, a_{t+1}), s_{t+2}$. This can be done by either directly exploiting the knowledge of the environment (Huang et al., 2019; Singh & Zheng, 2021; Xu & Zhang; Singh et al., 2021; 2022)(they used the rendering function being used in the environment to train agents) or learning the dynamics model ($f_{trans} : \mathcal{S} \times \mathcal{A} \to \mathcal{S}$) and the reward model ($f_{rew} : \mathcal{S} \times \mathcal{A} \to \mathbb{R}$). Thus, the policy can be formulated as follows

$$\pi(s) = \text{argmax}_{a \in \mathcal{A}}[f_{rew}(s_t, a_t) + \gamma * V(f_{trans}(s_t, a_t))]$$

This formulation shows that the value-based method delegates the task of imagining the next state given the current step to the dynamics and the reward models. With this formulation, Szubert & Jaśkowski (2014) proposed the afterstate value-based method and empirically showed its capability.

## 3.2 AFTERSTATE RL FOR CONTINUOUS CONTROL

Discrete policies cannot be directly scaled to continuous tasks. Yet, there is a number of work addressing the continuous RL tasks in the community. Thus, we take DDPG (Lillicrap et al., 2015) as an example to show how to realise the afterstate version of the critic in the actor-critic algorithms.

In the formulation of DDPG, let $(s_i, a_i, r_i, s_{i+1})$ a tuple that is randomly sampled from a replay buffer and $Q(s, a|\theta^Q)$ and $Q'(s, a|\theta^{Q'})$ denote the critic network and the target critic network with the corresponding weights, respectively, and $\mu(s|\theta^\mu)$ be the actor with the corresponding weights.

First, we derive the aftertaste modification to the critic's temporal difference update with the state-value network $V(s|\theta^V)$ and the target state-value network $V'(s|\theta^{V'})$ by plugging Eq.1 in;

$$L = \frac{1}{N} \sum_t \Big( \underbrace{r(s_t, a_t) + \gamma * (r_{t+2} + V'(s_{t+2}|\theta^{V'}))}_{r(s_t,a_t) + \gamma Q'(s_{t+1}, a_{t+1}|\theta^{Q'})} - \big( \underbrace{r(s_t, a_t) + \gamma * V(s_{t+1}|\theta^V)}_{Q(s_t, a_t|\theta^Q)} \big) \Big)^2 \quad (2)$$

Similarly, we can define the actor loss using the sampled policy gradient;

$$\nabla_{\theta^\mu} J \approx \frac{1}{N} \sum_t \nabla_a \big( \underbrace{f_{rew}(s, a) + \gamma * V(s')}_{Q(s,a|\theta^Q)} \big)|_{s=s_t, a=\mu(s_t), s'=f_{trans}(s,a)} \nabla_{\theta^\mu} \mu(s|\theta^\mu)|_{s=s_t} \quad (3)$$

As the aftertaste is a simple plug-in framework, all the other parts of the algorithm (Algorithm 1 of (Lillicrap et al., 2015)) of DDPG remain the same.

The above formulation requires the dynamics model ($f_{trans}$) and the reward model ($f_{rew}$). In this work, we do not assume the knowledge of the environment nor offline dataset to pretrain such models. We use the model-based framework to train those models throughout training. The dynamics model is trained to reconstruct the next state $s_{t+1}$ given the pair of $(s_t, a_t)$ and the reward model is trained to estimate the reward at the time-step. The details of the dynamics and the reward models can be found in Sec.4. And the final algorithm, *Model-based Afterstate DDPG*, is described in Algorithm 1. The important design choices have been explored in Sec.5.3.

---

**Algorithm 1** Model-based Afterstate DDPG

---
1: **while** not converged **do**
2:      **for** update-step $c = 1, \cdots C$ **do**
3:          *// Dynamics and Reward learning*
4:          Draw a batch of data samples $\{(a_t, o_t, r_t)\}_{t=k}^{k+L}$ from the replay buffer
5:          Update $f_{trans}$ and $f_{rew}$ to minimize the predefined loss functions
6:
7:          *// Policy update*
8:          Update the critic by minimizing the loss Eq.2
9:          Update the actor by the critic Eq.3
10:      **end for**
11:
12:      *// Environment Interaction*
13:      **for** $t = 1, \cdots T$ **do**
14:          Select action $a \sim \pi(s) + \epsilon$ where $\epsilon \sim \mathcal{N}(0, \sigma)$
15:          Observe reward $r$ and new state $s'$
16:          Store transition tuple $(s, a, r, s')$ in the replay buffer
17:      **end for**
18: **end for**

---

## 4 ENVIRONMENTS / SIMULATORS

To illustrate the heaviness of learning the action utility on the environment and the effectiveness of our method of approaching the issue, we evaluate our method on the following three simulators; (1) Toy simulator of Pendulum where the state transition is predictable, (2) More advanced physics simulators where the state transition is obscure, and (3) Painting on the canvas where the next state is very difficult to predict due to the immense freedom of creativity. Figure 1 provides an overview of the tasks.

### 4.1 PENDULUM AND MUJOCO SIMULATORS

For physical control tasks, we used the suite of MuJoCo continuous control tasks (Todorov et al., 2012), interfaced through OpenAI Gym (Brockman et al., 2016).

**Dynamics and Reward Models**: To focus on investigating the effectiveness of the afterstate framework, we let agents directly observe the internal state of robots as working on the partial observation (POMDP) comes with additional complexity, e.g., dealing with the vision-based observation that requires an extra network to process images. The dynamics model in these asks is kept simple such that we use the autoencoder (Bank et al., 2020) with encoder and decoder being a 2-layer MLP with ReLU (Agarap, 2018) activation function in the middle. Thus, it minimises the following mean squared loss (MSE); $\frac{1}{N} \sum_{i=1}^{N} (s_i - \hat{s}_i)^2$ where $s$ represents the state sampled from the replay buffer. Similarly, the reward model is an autoencoder with the same architecture as the dynamics model except the output now being one-dimensional as the reward in these tasks is scalar. And the loss function is also the same MSE as above with the state being the reward.

### 4.2 PAINTGYM

The task here is to reconstruct the target image by painting on the canvas with the given brush. Given a target image $I$ and an empty canvas $C_0$, the agent learns to find a sequence of strokes $(a_0, a_1, \cdots, a_{n-1})$ such that rendering a stroke $a_t$ on $C_t$ returns $C_{t+1}$. Through a sequence of interactions between the agent and canvas, the agent needs to produce the final canvas $C_n$ that is as visually similar to the target image $I$ as possible. This task has been modelled as a Markov Decision Process as follows (Huang et al., 2019; Singh & Zheng, 2021);

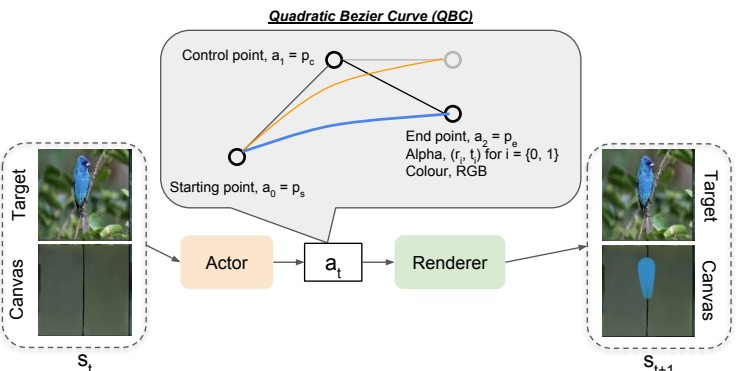

Figure 1: Task description of PaintGym.

**MDP formulation**: For *state and State Transition*, the state consists of three components $s_t = (C_t, I_t, t)$; the canvas, the target image, and the timestep index that conveys the information about the remaining number of steps to the agent. The transition function is the rendering of the stroke on the canvas. For *action*, we adopt the quadratic Bezier curve (QBC) as stroke representation to simulate the effects of brushes. The shape of the Bezier curve is specified by three control points in the 2D coordinate space of the canvas. Formally, the stroke is defined as the following;

$$a_t = (x_0, y_0, x_1, y_1, x_2, y_2, r_0, t_0, r_1, t_1, R, G, B) \in \mathbb{R}^{13}$$

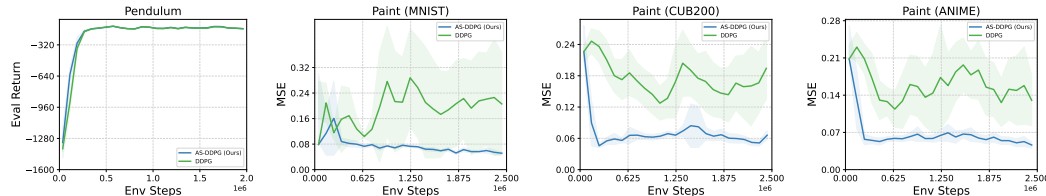

Figure 2: Results on Pendulum and PaintGym aggregated over 5 to 7 random seeds.

| Task | AS-SAC (Ours) | SAC | TD3 | DDPG |
|---|---|---|---|---|
| Ant | **3778.6 ± 1107.6** | 3722.6 ± 926.5 | 3326.2 ± 1110.9 | 468.7 ± 205.1 |
| Cheetah | **10012.8 ± 960.3** | 8478.4 ± 1704.1 | 9585.8 ± 1304.2 | 7932.9 ± 1857.4 |
| Hopper | 2929.4 ± 488.6 | 1491.1 ± 1517.6 | **3201.2 ± 237.7** | 665.2 ± 311.8 |
| Humanoid | 2325.1 ± 1758.5 | 2325.2 ± 2235.2 | **4593.7 ± 1739.4** | 1908.9 ± 199.6 |
| Pusher | -476.0 ± 74.1 | **-390.3 ± 33.4** | -401.2 ± 15.9 | -465.1 ± 85.3 |
| Reacher | **-41.5 ± 10.3** | -52.0 ± 8.8 | -55.6 ± 11.8 | -91.4 ± 54.8 |
| Swimmer | 56.0 ± 17.8 | 48.5 ± 1.7 | 47.1 ± 3.3 | **136.1 ± 18.6** |
| Walker2D | **3641.4 ± 938.5** | 947.4 ± 756.8 | 1856.0 ± 969.4 | 1392.9 ± 559.2 |

Table 1: Episode return of evaluation episodes on Mujoco tasks. The numbers are in the format of the mean ± STD. Results are aggregated over 4 to 7 random seeds. Bold numbers indicate the most performant agent in each task.

where $(x_i, y_i)$ is the coordinates of the $i$-th control points of the QBC, and $(r_i, t_i)$ controls the thickness and transparency of the two endpoints of the QBC respectively. Finally, for *reward*, $r(s_t, a_t) = L_t - L_{t+1}$ is designed to encourage the agent to take action that can paint towards the target image as similar as possible. Prior works Huang et al. (2019); Singh & Zheng (2021) used the discriminator in WGAN Arjovsky et al. (2017) to give out rewards. Formally, the reward is obtained by $D(C_{t+1}, I) - D(C_t, I)$ where $D$ is the discriminator.

**Dataset**: We employed the following three datasets: (I) *MNIST* (LeCun, 1998) is a classic image classification dataset of hand-written digits consisting of 60,000 training images and 10,000 validation images. Each example is a grayscale image of 28 × 28 pixels. (II) *CUB-200 Birds* (Wah et al., 2011) is a large-scale bird-image dataset of 200 bird species commonly used in benchmarking visual classification models. The dataset is considered as challenging as the images present high variation in object background as well as scale, position and the relative saliency of the foreground bird. (III) *Anime dataset* (Chen et al., 2018) is constructed by sampling images from the videos of stories from Hayao Miyazaki. Since the dataset was not publicly available, we have decided to use another source[1] for this dataset. The dataset consists of 1752 images that have been taken from "The Wind Rises" by Hayao Miyazaki.

**Dynamics and Reward Models**: Similar to the ones in Mujoco tasks, we decided to use a conditional Variational Auto-Encoder (VAE) (Sohn et al., 2015). Thus, the model takes as input $s_t = C_t$ the canvas (we do not use other state components as they are contributing little to the state transition) and $a_t$ being a brush stroke and learns to reconstruct the next canvas after executing the brush stroke. The loss formulation follows the conventional VAE such that a combination of the generation loss and the Kullback–Leibler divergence loss.

## 5 EXPERIMENTS

To examine the capability of the aftertaste framework, we design experiments to answer the following questions; (1) If the afterstate framework is effective? (2) What is the effect of the learnt dynamics and the reward models in Actor-critic algorithms? (3) What are important design choices? (4) Is the effectiveness of the afterstate framework scale to other actor-critic algorithms?

---

[1] https://github.com/TachibanaYoshino/AnimeGANv2

## 5.1 EFFECTIVENESS OF AS DDPG

We evaluate our proposition against the state of the art actor-critic algorithms, DDPG (Lillicrap et al., 2015), TD3 (Fujimoto et al., 2018), and SAC (Haarnoja et al., 2018), to validate the effectiveness of the afterstate framework. Fig. 4.2 and Table. 4.2 show the comparison of our agent to the baselines on the three continuous control environments and below we discuss the individual results.

**Pendulum**: Fig.4.2 shows that both our method and the baseline DDPG perform on par at the optimal level in this toy task. This is expected as this task comes with an easily predictable transition that is based on the action taken. We observed that our method shows a faster convergence than DDPG.

**PaintGym**: Fig.4.2 shows that in MNIST our method outperformed the baselines across all three datasets. In CUB200 and ANIME datasets, the baselines show the sign of learning yet the convergence speed is much slower than the one of our method. In particular, the handwritten characters in MNIST images have a completely black background and this requires more crispy painting compared to the images in the other datasets in the task. Thus, learning in MNIST can be more challenging than the other datasets and we can see that all agents perform worse than the levels that they achieved in other datasets.

Fig.3 shows the canvases at the final step in the evaluation episode as qualitative outcomes of agent training. In MNIST images, our agents were able to paint clearly, especially the edge of handwritten characters were reproduced. In more composite images from CUB200 and Anime datasets, the agent was able to follow the colours in the target images but the boundary of objects, e.g., birds, people, or some background objects in scenery.

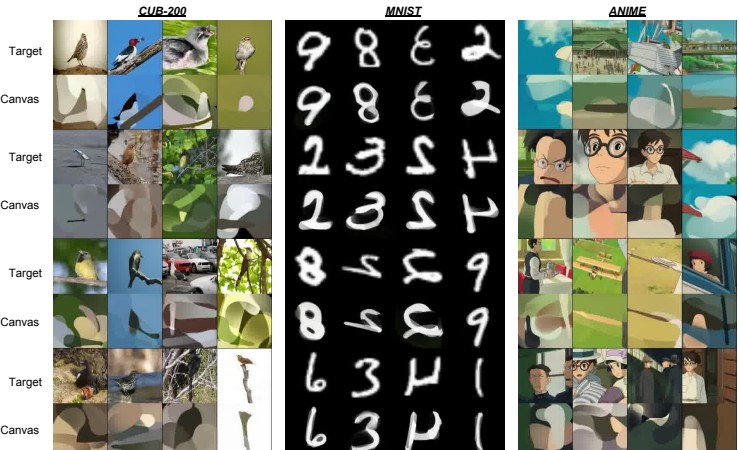

Figure 3: Qualitative performance painting after training. The first row is the target images and the second row is the painted canvases. The same pair of roles apply to the following rows as well.

**Mujoco**: Table.4.2 shows the comparison of our method to the baselines across different Mujoco tasks. In the tasks of *Cheetah, Reacher, Swimmer, and Walker2D*, our method was able to outperform all three baselines. But, in *Ant, Humanoid, and Pusher*, our agents fell short of the baseline performance. In the following section, we further analysed those results from different perspectives.

In summary, as the effectiveness of the afterstate framework, we observed faster convergence in Pendulum and PaintGym and better performance in PaintGym and some of Mujoco tasks.

## 5.2 QUALITATIVE ANALYSIS

We performed two different analyses to examine the effect of the afterstate framework.

**Learnt value function**: We analyse how the learnt value functions between ours and the baseline are different by evaluating Q-values on all the discredited actions of the 1-dimensional action space ($[-2, 2]$) of the Pendulum simulator. In the initial state of the Pendulum task, the direction of swinging the pendulum does not matter as we can just swing in the opposite direction subsequently to

utilise the inertia. Fig.4 shows that our learnt value function (Left) models this well such that it puts a high value to both edges of the action space whereas the one of the baseline (Right) has learnt to prefer the right direction. We have to note that as Fig.4.2 shows, in the end, both methods achieve the same optimality but ours had the faster convergence.

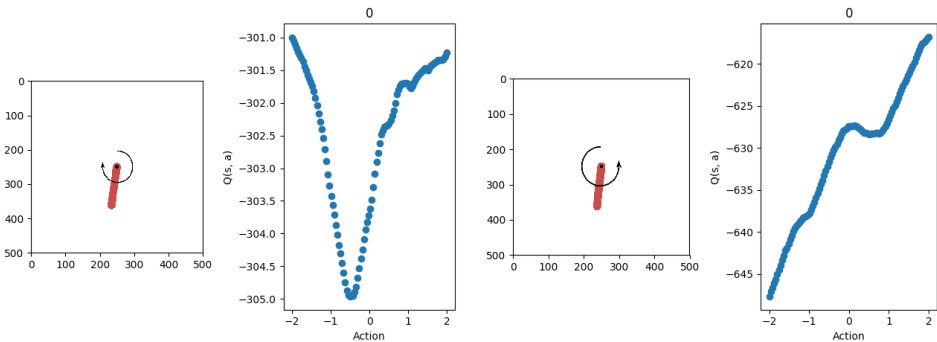

Figure 4: *(Left) AS-SAC (Ours) and (Right) Baseline SAC*: Screenshot of Pendulum simulation along with the plot of Q-values evaluated at 200 discretised actions at the initial state.

### 5.3 DESIGN CHOICES

We performed a series of comparisons to investigate the performance differences with regard to the design choices in Table.5.3. First, let us describe our variants in the following;

**Latent-state or full-State for afterstate (AS-SAC-Latent)**: Recent works of the model-based RL (Lee et al., 2020; Hafner et al., 2019b) show the promising capability of planning in the latent space. So, we hypothesize that the latent state space can benefit in the afterstate framework by offering lower dimensionality as well as smooth state transition capturing the underlying dynamics. So, we implemented the *AS-SAC-Latent* that employs the critic which takes as input the following; $(s_t, z_{t+1} = f_{\text{trans-enc}}(s_t, a_t))$, thus, the critic can focus on learning the state value in the latent state space.

**With or Without Current state as input (AS-SAC-SingleInput)**: In principle, the afterstate based critic relies on the next state in value computation. However, we hypothesize that adding the previous state as input can help the critic to induce the action taken between the current and the predicted next states. So, we implemented the *AS-SAC-SingleInput* that does not input the current state as input and purely relies on the predicted next state in value computation.

**With or Without Reward in Actor's update (AS-SAC-AR)**: In the actor's loss formulation (Eq.3), we hypothesize that the actor can have choices whether to maximise the predicted immediate reward from the reward model or follow the critic. So, we implemented the *AS-SAC-AR* that uses the predicted reward in the actor's update.

**Result**: We observed that missing the current state information harmed the performance of the agent and the immediate reward signal helps the actor learn. Surprisingly, the latent afterstate did not seem to help in our experiments.

| Task | AS-SAC (Ours) | AS-SAC-SingleInput | AS-SAC-Latent | AS-SAC-AR |
|---|---|---|---|---|
| Ant | $3778.6 \pm 1107.6$ | $-629.9 \pm 1642.9$ | $1283.1 \pm 2869.3$ | $\mathbf{4396.1 \pm 1430.2}$ |
| Cheetah | $\mathbf{10012.8 \pm 960.3}$ | $6221.2 \pm 938.0$ | $6221.2 \pm 938.0$ | $9430.8 \pm 1917.2$ |
| Hopper | $\mathbf{2929.4 \pm 488.6}$ | $1220.6 \pm 371.8$ | $1330.9 \pm 609.6$ | $2925.8 \pm 541.1$ |
| Humanoid | $2325.1 \pm 1758.5$ | $150.3 \pm 104.6$ | $371.7 \pm 111.1$ | $\mathbf{3388.6 \pm 1843.1}$ |
| Swimmer | $56.0 \pm 17.8$ | $41.7 \pm 9.7$ | $\mathbf{67.2 \pm 38.3}$ | $49.5 \pm 1.1$ |
| Walker2D | $\mathbf{3641.4 \pm 938.5}$ | $264.1 \pm 134.2$ | $811.3 \pm 621.8$ | $3533.1 \pm 739.1$ |

Table 2: Result of design choice experiments on different Mujoco tasks aggregated over 4 to 7 random seeds.

## 5.4 CONSISTENCY CHECK

Afterstate is a rather general framework so we investigated its applicability in all the baselines to see if it improves the performance as we observed in Fig.4.2 and Table.4.2. Table.3 shows the results of the application of the afterstate framework to the baselines and we observe that in many of Mujoco tasks, the aftertaste framework helps in those agents. We also observe that there are some tasks (Humanoid, Pusher, Reacher) afterstate-integrated agents consistently perform poorly. A thorough investigation of these tasks can be an interesting future direction.

| Task | SAC | AS-SAC-State | TD3 | AS-TD3-State | DDPG | AS-DDPG-State |
|---|---|---|---|---|---|---|
| Ant | $3722.6 \pm 926.5$ | $\mathbf{3778.6 \pm 1107.6}$ | $3326.2 \pm 1110.9$ | $\mathbf{4473.4 \pm 837.4}$ | $468.7 \pm 205.1$ | $\mathbf{1863.1 \pm 284.6}$ |
| Cheetah | $8478.4 \pm 1704.1$ | $\mathbf{10012.8 \pm 960.3}$ | $9585.8 \pm 1304.2$ | $8971.5 \pm 1715.3$ | $7932.9 \pm 1857.4$ | $\mathbf{9520.1 \pm 1578.0}$ |
| Hopper | $1491.1 \pm 1517.6$ | $\mathbf{2929.4 \pm 488.6}$ | $3201.2 \pm 237.7$ | $\mathbf{3223.2 \pm 370.7}$ | $665.2 \pm 311.8$ | $\mathbf{1187.1 \pm 270.1}$ |
| Humanoid | $\mathbf{2325.2 \pm 2235.2}$ | $2325.1 \pm 1758.5$ | $\mathbf{4593.7 \pm 1739.4}$ | $2750.4 \pm 2689.2$ | $\mathbf{1908.9 \pm 199.6}$ | $854.2 \pm 462.4$ |
| Pusher | $\mathbf{-390.3 \pm 33.4}$ | $-476.0 \pm 74.1$ | $\mathbf{-401.2 \pm 15.9}$ | $-523.5 \pm 67.7$ | $\mathbf{-465.1 \pm 85.3}$ | $-566.6 \pm 167.6$ |
| Reacher | $-52.0 \pm 8.8$ | $\mathbf{-41.5 \pm 10.3}$ | $\mathbf{-55.6 \pm 11.8}$ | $-69.7 \pm 21.6$ | $\mathbf{-91.4 \pm 54.8}$ | $-117.1 \pm 62.1$ |
| Swimmer | $48.5 \pm 1.7$ | $\mathbf{56.0 \pm 17.8}$ | $47.1 \pm 3.3$ | $\mathbf{50.1 \pm 1.1}$ | $136.1 \pm 18.6$ | $\mathbf{143.5 \pm 24.2}$ |
| Walker2D | $947.4 \pm 756.8$ | $\mathbf{3641.4 \pm 938.5}$ | $1856.0 \pm 969.4$ | $\mathbf{3899.7 \pm 521.0}$ | $\mathbf{1392.9 \pm 559.2}$ | $299.0 \pm 53.7$ |

Table 3: Bold numbers indicate a greater value within the same class of algorithm.

## 5.5 LIMITATION: BENCHMARKING WITH MODEL-BASED RL STATE-OF-THE-ART

To analyse our method more, we have employed the State-of-the-Art model-based RL method, Model-Based Policy Optimization (MBPO; (Janner et al., 2019)), to benchmark on Mujoco tasks. Fig. 5.5 shows that the performances of *SAC*, MBPO on SAC (*MBPO*), and SAC with Afterstate extension (*AS-SAC*). We followed all the hyper-parameter settings in the original work of (Janner et al., 2019) and implemented our method based on the public repository[2].
*A word of warning*: the public implementation of MBPO relies on a different version of Mujoco and a different environment configuration where the agent only interact with one environment to emphasise the sample efficency whereas all our experiments (e.g., Table.4.2) are conducted on the *vector* environment that agents interact with a batch of environments. Thus, these factors led to the difference in the results.

In all tasks, our method outperforms SAC, yet, in the tasks of *Ant, Cheetah, and Hopper*, we observed the better sample efficiency of MBPO. Yet, in most tasks (*Cheetah, Hopper, and Walker2d*), our method was able to reach the similar optimality. In particular in *Walker2D*, we were able to beat MBPO. Although we omitted from the plots, interestingly, we empirically observed that the combination of MBPO and the aftertaste framework did not offer improvements. This can be partly because, in the afterstate framework, we compute $r(s_{t+1}, a_{t+1}), s_{t+2}$ (See Sec.3.1) based on those hallucinated samples that can incorporate some errors and *95 per cent* (Appendix C of (Janner et al., 2019)) of samples used to update the agent are the model-generated ones. Thus, this can cause compounding errors. Yet, further analysis is needed to conclude.

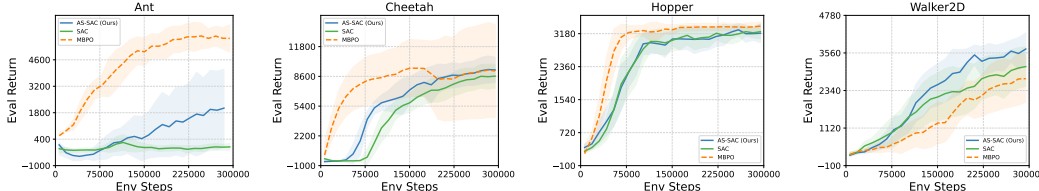

Figure 5: Comparison of our method to model-based RL method on Mujoco aggregated over 4 to 10 random seeds.

## 6 RELEVANT WORKS

**Afterstate in RL**: The concept has been originally discussed in Sec.6 of (Sutton & Barto, 2018) as *Afterstate* or Powell (Powell, 2007) formulated slightly differently as *Poststate*. In boardgames,

---

[2]https://github.com/Xingyu-Lin/mbpo_pytorch

Szubert & Jaśkowski (2014) concretely formulated the concept in the value-based method and empirically showed that in the aftertaste version of temporal difference (TD) learning outperformed vanilla TD learning. Some works consider the larger scale setting in discrete tasks as well the boardgames (Matsuzaki, 2021; Antonoglou et al., 2021) and Power-supplying network management (Yoon et al., 2021). Recently, in the computer vision domain, there is a line of work considering the application of DDPG Lillicrap et al. (2015) to paint a picture on the canvas. This work pretrain the rendering function that draws the agent's action (a brush stroke) on the canvas to generate the next canvas. This rendering function is actually being used in the environment thus agents have access to the privileged information about tasks whereas we do not assume this and train the dynamics model over the course of training.

**Equivalance in tasks**: Finding the equivalence in the task space has been proven to reduce the problem into smaller MDP in which we can efficiently learn (Ravindran & Barto, 2001; 2004). MDP Homomorphisms has been defined as a mapping between MDPs and the Bisimulation metrics (Ravindran & Barto, 2001) quantifies the behavioural similarity between states in the discrete MDPs (Van der Pol et al., 2020) and the continuous MDPs (Rezaei-Shoshtari et al., 2022) based on the reward. These approaches essentially focus on how to group states that inherit the similarity in the task. Yet, our work and afterstate framework focus on representing the action by the next state. Thus, this can be an interesting direction for future research.

**Differentiable Simulators** One can use the finite difference methods (e.g., explicit or implicit Euler methods) to estimate the gradient flowing through the physics simulators. Yet, these methods come with a higher-order error term. Recently, some works proposed incorporating a differentiable simulator into the algorithm (Pretrained Renderer; Huang et al. (2019); Mellor et al. (2019) or Physics simulators Wiedemann et al. (2022)) to directly optimize the agent towards the goal of the task. Yet, we noticed that model-based RL offers the flexibility to learn various dynamics thus we decided to employ it in the afterstate framework.

**Model-based RL** Training to approximate the environment is known to offer advantages, e.g., better sample efficiency or the capability to simulate the near future to make a better immediate decision. Stochastic Latent Actor-Critic (Lee et al., 2020) proposed to train the actor-critic method in the latent space constructed by the state space model approximating the environment. Model Predictive Control ( (Hafner et al., 2019a), Dreamer-v2 (Hafner et al., 2020), and Dreamer-v3 (Hafner et al., 2023)) has been used to conduct the planning over a certain horizon and work out the best immediate action. Recently, Model-Based Policy Optimization (MBPO; (Janner et al., 2019)) proposed to roll out the agent with the state/reward models and store the samples in the replay buffer to update the agent. Our focus, however, directs towards incorporating the state/reward models in the update of the agent itself rather than utilising them in decision-making or hallucinating the samples. To examine our method more, we conducted a comparison of our method to MBPO in the experiments.

## 7 CONCLUSION

We present the afterstate framework combined with the model-based RL in the actor-critic algorithms for continuous tasks. Our method leverages the environment models (i.e., the dynamics and the reward models) in learning the future value for the critic and updating the actor. We demonstrate that learning the knowledge of the environment is helpful for optimal decision-making in three continuous tasks.

## 8 REPRODUCIBILITY STATEMENT

To guarantee that our findings can be replicated, we've included our code in the supplementary materials. This code encompasses all the necessary environments and baseline methods mentioned in our paper. Additionally, you can find information about the hyperparameters for our experimental setting in Appendix D.

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

## A  APPENDIX

## B  OPTIMALITY CHECK IN TOY PENDULUM

To investigate the optimality of the learnt agent in the toy experiment, we implemented the ground-truth dynamics and reward models in the Pendulum task to train our agent. In the plot, *AS-SAC (GT)* is the one implemented in this section such that SAC with the ground truth dynamics and the reward models and *AS-SAC (Ours), SAC* are from the main experiment (Fig.2) of the main text. Fig.6 shows that *Ours* and *GT* have converged to almost the same optimality, meaning that our agent was able to learn as well as the aftertaste framework with ground-truth access.

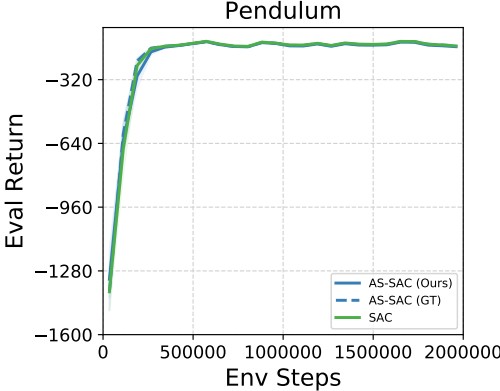

Figure 6: Result of comparison with ground-truth dynamics and reward models in Pendulum task. Aggregated over 4 random seeds.

## C   DQN ON HIGHLY STOCHASTIC ENVIRONMENT

Real-world tasks naturally come with certain stochasticity involved. Yet, the afterstate framework assumes the correspondence of action and the next state, in other words, it assumes the 1-to-1 mapping between the action and the next state. So, we decided to deploy our agent to a stochastic environment to examine the heaviness of this assumption. Finally, as a representative of real-world stochastic tasks, we chose to use the recommender systems that exhibit the stochasticity nature of user behaviour (e.g., a user might consider the movie "Titanic 1997" as an epic romance movie but another user can consider it as a disaster movie ) in the task.

### C.1   RECSIM

The simulated RecSys environment RecSim (Ie et al., 2019; Jain et al., 2021), requires an agent to select an item that matches the user's interest from a large item set. In this environment, users have been implemented with a dynamically changing preference upon clicks. Thus, the task for agents is to infer this changing user preference from user clicks and recommend the most relevant item to maximize the total number of clicks in the episode. The details of the MDP setting can be found in (Ie et al., 2019; Jain et al., 2021) but let us briefly describe the important components.

**State and Transition**: The user interest embedding ($e_u \in \mathbb{R}^n$ where $n$ denotes the number of categories of items) represents the user interest in categories that change over time as the user consumes items upon click. Upon the user clicks an item with the item representation ($e_i \in \mathbb{R}^n$; the same $n$), the user interest embedding updates as follows;

$$\Delta(e_u) = (-y|e_u| + y) \cdot (1 - e_u), \text{ for } y \in [0, 1]$$
$$e_i \leftarrow e_u + \Delta(e_u) \text{ with probability}[e_u^T e_i + 1]/2$$
$$e_u \leftarrow e_u - \Delta(e_u) \text{ with probability}[1 - e_u^T e_i]/2$$

**Action**: The set of recommendable items is given to the agent. Thus, the agent has to find the most relevant item to a user given this item set.

**Reward**: The reward is simulated user feedback (i.e., clicks). The user model (Ie et al., 2019) stochastically skips or clicks the recommended item based on the present preference ($e_u$) by computing the following score;

$$\text{score}_{item} = \langle e_u, e_i \rangle$$
$$p_{item} = \frac{e^{score_{item}}}{e^{s_{item}} + e^{score_{skip}}}$$
$$p_{skip} = \frac{e^{score_{skip}}}{e^{s_{item}} + e^{score_{skip}}}$$

where $\langle \cdot, \cdot \rangle$ is the dot product notation and $score_{skip}$ is a task hyper-parameter. Finally, the user model stochastically selects either click(reward=1) or skip(reward=0) based on the categorical distribution on $[p_{item}, p_{skip}]$.

**Item Representations**: Following Jain et al. (2021), we implement continuous item representations sampled from a Gaussian Mixture Model (GMM) with centres around each item category. Note that, in this work, we do not use the sub-category in the category system.

**Dynamics and Reward Models**: To focus on investigating the effectiveness of the afterstate framework, we employ the same simple DNNs for the dynamics and the reward models as Mujoco tasks in the main text. The same autoencoders (Bank et al., 2020) with encoder and decoder being a 2-layer MLP with ReLU (Agarap, 2018) activation function in the middle are used for the dynamics and the reward models. The same loss formulation as Mujoco tasks has been used.

## C.2 AGENTS

To implement the afterstate framework in the agents for RecSim, we chose to use DQN (Mnih et al., 2015) that has been augmented with the item representation as input Jain et al. (2020; 2021); Ie et al. (2019) such that DQN takes as input state (user representation $e_u \in \mathbb{R}^n$) and action (item representation $e_i \in \mathbb{R}^n$). So in the experiments, we compared the following agents; (i) *AS-DQN*: We integrated the afterstate framework in DQN by expanding the Q-value in the decision-making like (Szubert & Jaśkowski, 2014; Matsuzaki, 2021);

$$\pi(s) = \mathrm{argmax}_{a \in \mathcal{A}} f_{rew}(s, a) + V(f_{trans}(s, a))$$

The temporal difference loss (TD-loss) is the same as the critic formulated in the main text. (ii) *AS-DQN-Latent*: To see if the dimension reduction of the latent space helps, we added this variation.

## C.3 RESULT

Recall that we hypothesized that the afterstate framework suffers in a stochastic state transition, like recommender systems, in the task. First of all, Fig.7 (Left) shows that AS-DQN fails to learn to solve the task whereas the baseline DQN learns to perform well. To investigate the cause, we checked the following factors; (i) Dynamics model training went well thus the loss curve looked almost the same as the ones in the main experiment of Sec.5 so we do not show here, (ii) Reward model training suffers from the stochasticity of the user model. This can be seen in Fig.7 (Middle), (iii) Plot of the standard derivation of Q-values (Right-most plot of Fig.7) shows that our variants fail to differentiate actions as the plot shows that our variants have assigned similar Q-values to all actions whereas the baseline DQN increased the standard derivation, which implies that the baseline DQN learns to distinguish actions.

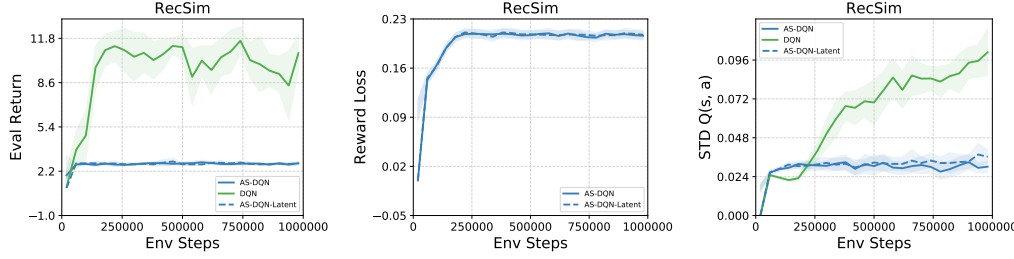

Figure 7: (Left) Comparisons of our variants to the baseline DQN regarding Episode return, (Middle) Reward loss curve of our variants, (Right) Standard Deviation of Q-values in decision-making of our variants. All results are aggregated over 4 random seeds.

## D IMPLEMENTATION DETAILS

We used PyTorch (Paszke et al., 2019) for our implementation, and the experiments were primarily conducted on workstations with either NVIDIA P40 or V100 GPUs on NAVER Smart Machine

Learning platform (NSML) (Kim et al., 2018). Each experiment seed takes about 1 hour for Mujoco and Pendulum tasks and 17 hours for PaintGym to converge. We use the Weights & Biases tool (Biewald, 2020) for logging and tracking experiment runs. All the environments were developed using the OpenAI Gym interface (Brockman et al., 2016).

## D.1 HYPERPARAMETERS

| Hyperparameter | Pendulum | MUJOCO | PaintGym | RecSim |
|---|---|---|---|---|
| **Environment** | | | | |
| Total timesteps | 1M | 10M | 2.5M | 1M |
| Parallel environments | 16 | 16 | 16 | 16 |
| max episode steps | 100 | 1000 | 50 | 20 |
| **RL Training** | | | | |
| Actor lr | 0.001 | 0.001 | 0.0001 | 0.0001 |
| Critic lr | 0.001 | 0.001 | 0.001 | 0.001 |
| Batch size | 256 | 256 | 256 | 256 |
| Buffer size | 500K | 500K | 500K | 500K |
| Critic gamma | 0.99 | 0.99 | 0.99 | 0.99 |
| Actor eval epsilon | 0 | 0 | 0 | 0 |
| Critic eval epsilon | 0 | 0 | 0 | 0 |

Table 4: Environment/Policy-specific Hyperparameters

## D.2 NETWORK ARCHITECTURES: PENDULUM AND MUJOCO TASKS

```
# Actor
Actor(
  (l1): Linear(in_features=dim-action, out_features=256, bias=True)
  (l2): Linear(in_features=256, out_features=256, bias=True)
  (mean_linear): Linear(in_features=256, out_features=8, bias=True)
  (log_std_linear): Linear(in_features=256, out_features=8, bias=True)
)
# Critic
dim = dim-state if if_afterstate else dim-state + dim-action
Critic(
  (l1): Linear(in_features=dim, out_features=256, bias=True)
  (l2): Linear(in_features=256, out_features=256, bias=True)
  (l3): Linear(in_features=256, out_features=1, bias=True)
  (l4): Linear(in_features=64, out_features=256, bias=True)
  (l5): Linear(in_features=256, out_features=256, bias=True)
  (l6): Linear(in_features=256, out_features=1, bias=True)
)

# State Model
Model(
  (_enc): Sequential(
    (0): Linear(in_features=dim-state + dim-action, out_features=256,
    bias=True)
    (1): ReLU()
    (2): Linear(in_features=256, out_features=32, bias=True)
  )
  (_dec): Sequential(
    (0): Linear(in_features=32, out_features=256, bias=True)
    (1): ReLU()
    (2): Linear(in_features=256, out_features=27, bias=True)
  )
  (criterion): MSELoss()
)
```

```
33
34  # Reward Model
35  Model(
36    (_enc): Sequential(
37      (0): Linear(in_features=dim-state + dim-action, out_features=256,
      bias=True)
38      (1): ReLU()
39      (2): Linear(in_features=256, out_features=32, bias=True)
40    )
41    (_dec): Sequential(
42      (0): Linear(in_features=32, out_features=256, bias=True)
43      (1): ReLU()
44      (2): Linear(in_features=256, out_features=1, bias=True)
45    )
46    (criterion): MSELoss()
47  )
```

### D.3  NETWORK ARCHITECTURES: PAINTGYM

We employed the same architecture for Actor, Critic, and the reward model as (Huang et al., 2019)[3] so let us describe our cVAE-based state model.

```
1   cVAE(
2     (encoder): Encoder(
3       (encode): Sequential(
4         (0): Conv2d(3, 16, kernel_size=(5, 5), stride=(1, 1))
5         (1): BatchNorm2d(16, eps=1e-05, momentum=0.1, affine=True,
      track_running_stats=True)
6         (2): ReLU(inplace=True)
7         (3): Conv2d(16, 32, kernel_size=(5, 5), stride=(1, 1))
8         (4): BatchNorm2d(32, eps=1e-05, momentum=0.1, affine=True,
      track_running_stats=True)
9         (5): ReLU(inplace=True)
10        (6): MaxPool2d(kernel_size=2, stride=2, padding=0, dilation=1,
      ceil_mode=False)
11        (7): Conv2d(32, 64, kernel_size=(3, 3), stride=(1, 1))
12        (8): BatchNorm2d(64, eps=1e-05, momentum=0.1, affine=True,
      track_running_stats=True)
13        (9): ReLU(inplace=True)
14        (10): Conv2d(64, 64, kernel_size=(3, 3), stride=(1, 1))
15        (11): BatchNorm2d(64, eps=1e-05, momentum=0.1, affine=True,
      track_running_stats=True)
16        (12): ReLU(inplace=True)
17        (13): MaxPool2d(kernel_size=2, stride=2, padding=0, dilation=1,
      ceil_mode=False)
18        (14): Flatten()
19        (15): MLP(
20          (mlp): Sequential(
21            (Linear_0): Linear(in_features=50176, out_features=256, bias=
      True)
22            (BatchNorm_0): BatchNorm1d(256, eps=1e-05, momentum=0.1, affine
      =True, track_running_stats=True)
23            (ReLU_0): ReLU(inplace=True)
24            (Linear_1): Linear(in_features=256, out_features=128, bias=True
      )
25            (BatchNorm_1): BatchNorm1d(128, eps=1e-05, momentum=0.1, affine
      =True, track_running_stats=True)
26            (ReLU_1): ReLU(inplace=True)
27          )
28        )
29      )
30      (calc_mean): MLP(
```

[3]See Appendix 7.1: https://arxiv.org/pdf/1903.04411.pdf

```
31       (mlp): Sequential(
32         (Linear_0): Linear(in_features=160, out_features=64, bias=True)
33         (BatchNorm_0): BatchNorm1d(64, eps=1e-05, momentum=0.1, affine=
    True, track_running_stats=True)
34         (ReLU_0): ReLU(inplace=True)
35         (Linear_1): Linear(in_features=64, out_features=32, bias=True)
36       )
37     )
38     (calc_logvar): MLP(
39       (mlp): Sequential(
40         (Linear_0): Linear(in_features=160, out_features=64, bias=True)
41         (BatchNorm_0): BatchNorm1d(64, eps=1e-05, momentum=0.1, affine=
    True, track_running_stats=True)
42         (ReLU_0): ReLU(inplace=True)
43         (Linear_1): Linear(in_features=64, out_features=32, bias=True)
44       )
45     )
46   )
47   (decoder): Decoder(
48     (decode): Sequential(
49       (0): MLP(
50         (mlp): Sequential(
51           (Linear_0): Linear(in_features=64, out_features=64, bias=True)
52           (BatchNorm_0): BatchNorm1d(64, eps=1e-05, momentum=0.1, affine=
    True, track_running_stats=True)
53           (ReLU_0): ReLU(inplace=True)
54           (Linear_1): Linear(in_features=64, out_features=128, bias=True)
55           (BatchNorm_1): BatchNorm1d(128, eps=1e-05, momentum=0.1, affine
    =True, track_running_stats=True)
56           (ReLU_1): ReLU(inplace=True)
57           (Linear_2): Linear(in_features=128, out_features=256, bias=True
    )
58           (BatchNorm_2): BatchNorm1d(256, eps=1e-05, momentum=0.1, affine
    =True, track_running_stats=True)
59           (ReLU_2): ReLU(inplace=True)
60           (Linear_3): Linear(in_features=256, out_features=49152, bias=
    True)
61         )
62       )
63       (1): Sigmoid()
64     )
65   )
66   (label_embedding): Sequential(
67     (0): Linear(in_features=65, out_features=32, bias=True)
68   )
69   (a2img_encoder): Sequential(
70     (0): Conv2d(67, 64, kernel_size=(1, 1), stride=(1, 1))
71     (1): TReLU()
72     (2): Conv2d(64, 64, kernel_size=(1, 1), stride=(1, 1))
73     (3): TReLU()
74     (4): Conv2d(64, 32, kernel_size=(1, 1), stride=(1, 1))
75     (5): TReLU()
76     (6): Flatten()
77     (7): Linear(in_features=131072, out_features=128, bias=True)
78     (8): TReLU()
79     (9): Linear(in_features=128, out_features=32, bias=True)
80   )
81   (_criterion): BCELoss()
82 )
```

## D.4  NETWORK ARCHITECTURES: RECSIM

```
1 # Note: DQN's input is 3D-tensor -> (batch-size x num-actions x dim-state
      + dim-action)
```

```
2 DQN(
3   (0): Linear(in_features=dim-state + dim-action, out_features=128, bias=
      True)
4   (1): ReLU()
5   (2): Linear(in_features=128, out_features=64, bias=True)
6   (3): ReLU()
7   (4): Linear(in_features=64, out_features=1, bias=True)   # Output Q-
      values for each action: batch-size x num-actions
8 )
```

