# OpenReview forum: "Afterstate Reinforcement Learning for Continuous Control"
_ICLR.cc/2024/Conference — ICLR 2024 Conference Withdrawn Submission_

### Official Review · Reviewer_CtUq · 2023-10-31

**Soundness:** 2 fair
**Presentation:** 2 fair
**Contribution:** 1 poor
**Rating:** 3
**Confidence:** 5

**Summary:**

This paper aims to enhance the critic's imagination in Actor-Critic. The authors proposes two methods (1) learning the utility of action on the environment,  (2) learning the future consequence of the action. The authors verify the effective of their method in standard benchmark.

**Strengths:**

This paper proposes a simple method to improve the current RL methods in continuous control tasks. The work present the concept of the Afterstate framework and incorporate this into the critic in the actor-critic framework.

**Weaknesses:**

1. The presentation of the method can be highly improved. (1) The authors should provide detailed description of the motivation of using after-state framework. As it stands, this statement is very heuristic and not supported by any theoretical analysis or evidence.Therefore, this makes the paper significantly less convincing.

2. Experiments lack more advanced baseline algorithms, such as REDQ [1].

3. Although the author shows the performance of the algorithm in tables, it is more convincing to show the algorithm through curves.

4. The author has made minor changes to the classic Actor-Critic framework. From a technical point of view, the changes are very small and the novelty is not strong. Therefore, if the author can conduct a large number of experiments and beat the current best algorithms, it will strongly prove the importance of considering after-state in RL.

[1] Chen, Xinyue, et al. "Randomized ensembled double q-learning: Learning fast without a model." arXiv preprint arXiv:2101.05982 (2021).

**Questions:**

Please refer to the above weakness.

---

### Official Review · Reviewer_D12j · 2023-10-31

**Soundness:** 2 fair
**Presentation:** 2 fair
**Contribution:** 2 fair
**Rating:** 3
**Confidence:** 4

**Summary:**

The paper uses an actor-critic method that combines a model to
estimate the state-action value.  In particular Eq. (3) updates the
policy using an actor-critic method with the reward expanded out of
the value for one time-step using a learned model. While Eq. (2) defines
a TD-error based on a 2-time-step prediction.

The method is evaluated on control tasks like the pendulum and range of
MuJoCo tasks, as well as a painting task. They compare against SAC, MBPO
etc.

**Strengths:**

- A wide range of evaluations were considered.

**Weaknesses:**

- The idea of expanding out values using a learned model has been widely
studied in many past works, e.g., "Model-Based Value Expansion
for Efficient Model-Free Reinforcement Learning" (ICML2018, Feinberg et al.).
Also, methods like MuZero are also essentially performing value expansion
with a learned model. These works were not cited or discussed.
"Stochastic value gradients" is another related work that uses a learned model
to facilitate computation of the policy gradient.

- I am not confident in the evaluations. The benchmarking results for
SAC are weaker than other results that can be found online, e.g.:
https://github.com/ChenDRAG/mujoco-benchmark
https://spinningup.openai.com/en/latest/spinningup/bench.html

- I found that the clarity was not that great, and I think the manuscript could be polished
more.  Also I think some equations seem incorrect (see the bottom of the
weaknesses section for more details).


Comments:
The asterisk symbol * is not typically used to denote multiplication
in mathematics or machine learning publications.

Page 2: I think the temporal difference update for V seems wrong.
First, the left-hand side should have a time index for s.
Secondly, I believe the time index for r and for V(s_{t+2}) should
be shifted down by 1 step.

The notation on the bottom of page 2 for the policy also seems not accurate.
The argmax is over $a$, but the equation has $a_t$ with a time index.
Also, the left-hand side has $\pi(s)$, whereas the right-hand side has
a time index for $s_t$.

Similarly to the previous issue, equation 2 also seems inaccurate.
$Q_t = r_t + \gamma V(s_{t+1})$, but the left-hand side has the time-index for
$r$ wrong.

**Questions:**

Where did the code for the other algorithms such as
SAC come from, and why do the results not seem to match
the results in the existing benchmarks that I pointed out?

What is the relationship to value expansion methods?

---

### Official Review · Reviewer_Kkgd · 2023-11-04

**Soundness:** 3 good
**Presentation:** 2 fair
**Contribution:** 2 fair
**Rating:** 3
**Confidence:** 4

**Summary:**

The authors propose an RL algorithm that decouples learning transition dynamics of taking actions from estimating utility of the actions. This involves learning an explicit dynamics model for next state prediction, and training a value function on the next state (as opposed to learning Q functions).

**Strengths:**

While sample efficient RL is an important problem, I have concerns regarding the relation of the proposed method to existing approaches in the field, as detailed in the weaknesses section.

**Weaknesses:**

1. Relation to model-based RL approaches

The claim in the paper hinges on decoupling next state prediction (dynamics learning) from learning utility of actions. The authors then propose learning a separate dynamics model, which is used to optimize the control policy by rolling out the model for one step. Prior work [2,3] do many more rollout steps of the model for optimization, and show far greater sample efficiency than the proposed approach. It is unclear what is the benefit of using the dynamics model in the manner presented in the paper, as opposed to simulating samples from the dynamics model and using those to train the policy[1,2,3]. The authors do compare to MBPO [1], and show much worse sample efficiency. The argument for model-free methods is that dynamics might be hard to learn and unnecessary for the task, but since the presented method already learns a dynamics model, it is unclear why it is being used in the manner presented as opposed to the standard approach of simulating new samples. The paper also doesn't include comparisons to Dreamer and TD-MPC [2,3] which are more recent and much more efficient than MBPO. Furthermore from the experiments included, it seems the proposed approach does worse than its model-free counterparts on the challenging Humanoid environment, which indicates that the approach doesn't scale well to more complex environments.




[1] : Michael Janner, Justin Fu, Marvin Zhang, and Sergey Levine. When to trust your model: Modelbased policy optimization.
[2] : Danijar Hafner, Jurgis Pasukonis, Jimmy Ba, and Timothy Lillicrap. Mastering diverse domains through world models.
[3] : Hansen, Nicklas, Xiaolong Wang, and Hao Su. "Temporal difference learning for model predictive control."

**Questions:**

1. Why is it better to use the dynamics model in the manner presented, as opposed to simulating samples from the dynamics model as done in [1,2,3], given superior sample efficiency of the latter?

2. Please include learning curve comparisons to [2,3] on the standard dm-control benchmarks




[1] : Michael Janner, Justin Fu, Marvin Zhang, and Sergey Levine. When to trust your model: Modelbased policy optimization.
[2] : Danijar Hafner, Jurgis Pasukonis, Jimmy Ba, and Timothy Lillicrap. Mastering diverse domains through world models.
[3] : Hansen, Nicklas, Xiaolong Wang, and Hao Su. "Temporal difference learning for model predictive control."